# Optical Coherence Tomography Angiography in Macular Holes Autologous Retinal Transplant

**DOI:** 10.3390/jcm12062350

**Published:** 2023-03-17

**Authors:** Virgilio Morales-Canton, Daniela Meizner-Grezemkovsky, Pablo Baquero-Ospina, Nicolás Crim, Lihteh Wu

**Affiliations:** 1Retina Department, Asociación Para Evitar la Ceguera en México, “Hospital Dr. Luis Sanchez Bulnes”, IAP, Mexico City 04030, Mexico; 2Retina Department, Sanatorio del Salvador, Córdoba X5004, Argentina; 3Asociados de Macula, Vitreo y Retina de Costa Rica, San José 10102, Costa Rica; 4Department of Ophthalmology, Illinois Eye and Ear Infirmary, School of Medicine, University of Illinois, Chicago, IL 61801, USA

**Keywords:** autologous neurosensory retina transplant, macular hole surgery, OCT angiography, foveal avascular zone, vascular density, refractory macular hole

## Abstract

In this paper, we compare the post-operative macular microvascular parameters (vascular density and foveal avascular zone) in eyes with refractory macular hole (MH) that underwent pars plana vitrectomy and autologous retinal transplant (ART) with the fellow unoperated eye. We conducted a retrospective case control study of six consecutive patients who underwent pars plana vitrectomy and ART with at least six months of post-operative follow-up. Pre-operatively, all eyes underwent SD-OCT (Spectral Domain Optical Coherence Tomography) examination. Post-operative OCT-A analyses included vascular density (VD) and the foveal avascular zone (FAZ) area. Six patients with a mean age of 63.7 ± 14.3 years were included. The mean follow-up was 24 months (range 6–30 months). The pre-operative BCVA (best-corrected visual acuity) was 0.99 ± 0.46 logMAR and 1.02 ± 0.23 logMAR at the last post-operative visit (*p* = 1.00). The mean MH diameter was 966 ± 620 µm. VD in the MH group was 28.1 ± 7.3% compared to 20.2 ± 2.9% in the fellow eyes group (*p* < 0.05). The mean post-operative FAZ area in the MH group was 109.8 ± 114.6 mm^2^ compared to 41.5 ± 10.4 mm^2^ in the control group (*p* < 0.05). In all six eyes, MH closure was obtained. The post-operative visual acuity did not improve after ART. Eyes with a closed MH showed a bigger FAZ with a higher VD compared to the fellow healthy eye.

## 1. Introduction

In 1991, Kelly and Wendel [1] pioneered the modern treatment of full thickness macular holes (MHs) by recommending pars plana vitrectomy (PPV), peeling the posterior hyaloid, removal of any epiretinal membranes and gas tamponade. Since then, internal limiting membrane (ILM) removal has also been advocated by most vitreoretinal surgeons. Despite the improvements in technique and instrumentation over the past three decades, a small percentage of cases fail to close. Large MHs, defined as those with a diameter >400 μm [2] or >650 μm [3], and MHs that fail primary surgical treatment have lower rates of MH closure. Different techniques have been developed for these challenging cases.

Refractory MHs have a 70% closure rate compared to >90% in eyes with no prior surgery [4,5]. Over the years, several investigators have developed different techniques to improve the closure rates of refractory MHs. Alpatov and Malyshe [6] described a technique where the surgeon approximated the edges mechanically by holding and massaging the edges of the hole. They achieved a complete closure rate of 92%, but 72% of patients developed retinal pigment epitheliopathy, and the visual outcomes were not good. Charles and colleagues [7] performed an arcuate retinotomy temporal to the MH to increase retinal compliance and relieve tangential traction. They reported an 83% closure rate [7]. Michalewska [8] described the use of an inverted flap of ILM with a reported closure rate of 98%. In cases where no ILM is available, anterior and posterior lens capsular flaps [9]; human amniotic membrane [10]; and different hematic components, such as autologous blood and platelet-rich plasma [11,12], have all been used. These techniques rely on inducing glial cell proliferation, which bridges the macular defect, and this, in turn, enhances closure of the MH [8]. In 2016, Grewal and Mahmoud [13] described a novel surgical technique for refractory myopic MH with an autologous neurosensory retinal free flap or autologous retinal transplant (ART). A recent large case series using this technique reported a closure rate of 90% coupled to an improved best corrected visual acuity (BCVA) in 36% of the cases using this novel technique [14].

Several structural macular changes documented by SD-OCT have been described during the healing process [15,16]. Retinal vessels are a very important component of the retinal structure and very much involved in the healing of retinal diseases, including the closure process of MH [17]. Optical coherence tomography angiography (OCT-A) is a relatively new imaging modality that enables a depth resolved observation of the blood flow of each retinal capillary layer [18]. Baba and colleagues [19] showed that the foveal avascular zone (FAZ) area in the superficial capillary plexus decreased significantly following successful closure of the MH. This decrease in FAZ area suggests movement of the foveal tissue centripetally post-operatively [19]. Tabandeh [20] recently reported that, in eyes with giant MHs treated with ART, reperfusion and vascularization of the macular graft were achieved within 6 weeks of surgery. These findings suggest that autologous retinal grafts following ART are capable of surviving.

The purpose of the current study was to compare the post-operative macular vascular characteristics in eyes with an MH that underwent ART and the untreated fellow eye.

## 2. Materials and Methods

This retrospective case control study included 5 consecutive patients with a primary MH that did not close after a primary surgery and one additional patient with a very large MH. These patients underwent ART as described by Grewal and Mahmoud [13]. All patients were seen at the vitreoretinal service of the Asociación Para Evitar la Ceguera in Mexico City, Mexico, from October 2018 to March 2020. This was a retrospective study, so it was exempted from IRB review. Written informed consent was obtained from all the patients for vitrectomy surgery. This study adhered to the tenets of the Declaration of Helsinki.

### 2.1. Patient Eligibility and Exclusion Criteria

All patients with an MH that underwent surgical repair with ART were included in the study if they met the following criteria: (1) MH with a size of ≥800 µm and (2) no other possible causes for visual loss.

Patients were excluded if they had diabetic retinopathy, high myopia (spherical equivalent ≥6.0 diopters or axial length (AL) > 26 mm, glaucoma, anisometropia >2.0 diopters or other ocular pathology that could interfere with visual function.

### 2.2. Examination Procedures

At baseline and at each scheduled post-operative visit, each patient underwent a complete ophthalmic examination, including best corrected visual acuity (BCVA), slit lamp examination, intraocular pressure (IOP), fundus examination and spectral-domain optical coherence tomography (SD-OCT). A volume scan centered on the fovea was performed. The scans were reviewed, and manual corrections were performed in case of segmentation errors. Prior to surgery, all patients were staged by SD-OCT (Spectralis, Heidelberg Engineering, Heidelberg, Germany) examinations with measurements of the MH base diameter, the MH minimum linear dimension (width of MH at the narrowest point) and the MH apical inner opening.

Post-operative (6 months after ART) OCT-A imaging centered on the foveal area (6.0 × 6.0 mm–320 × 320) with a swept source device (Topcon, Triton, Topcon Healthcare) that uses split-spectrum amplitude decorrelation was obtained at 6 months post-operatively in both eyes of each patient. The FAZ area was manually measured on the internal edge in the density map, and the vascular densities (VD) in percentage (%) were measured with the device’s proprietary software (Topcon, Triton, Topcon Healthcare) and compared with those of the untreated fellow eyes. Factors related to macular vascular structural integrity and their associations with post-operative visual outcomes were evaluated.

### 2.3. Surgical Procedures

Five eyes underwent prior PPV for repair of a primary MH. One eye underwent a primary ART procedure. During the ART procedure, intravitreal triamcinolone (ATLC, Laboratorios Grin, Mexico City, Mexico) was used to verify that the posterior hyaloid had been peeled. Brilliant Blue G (DORC, Dutch Ophthalmic Instruments) was injected over the macular area to verify that there was no residual ILM at the edges of the MH. A neurosensory retina harvest site of approximately two-disc diameters was selected superior to the superotemporal arcade. Endocauterization was applied in a circular manner around it. Using chandelier illumination and a bimanual approach, the edge of the graft was held using ILM forceps and cut using vertical scissors (Figure 1). Perfluoro-n-octane (PFO) was instilled on the vitreous cavity, while taking care that the bubble covered the retina flap. The flap edges were gently flattened and stretched to lay over and cover the macular hole. Endolaser was applied around the donor area, and the vitreous cavity was filled with PFO. Three weeks after the first procedure, an air–PFO exchange was performed.

### 2.4. Statistical Analysis

The main outcomes measured were MH closure and the post-operative best corrected visual acuity (BCVA). MH closure was defined as flattening of the retinal detachment around the MH regardless of the configuration. Snellen visual acuities were transformed to the logMAR to facilitate the statistical analysis.

Statistical analysis was performed using Medical Statistics with R. A *p*-value ≤ 0.05 was considered statistically significant.

## 3. Results

Six patients (mean age, 63.7 ± 14.3 years; range, 35–74 years; IC95%, 48.5–78.7) were included. In the MH group, there were two right eyes. The mean follow-up was 24 months (range of 6–30 months). All patients had a mean of one prior to failed surgery (range of 0–2). One patient underwent a primary ART procedure. There were no intraoperative complications. One patient developed post-operative uveitis, most likely associated with the short-term PFO tamponade. This resolved following PFO removal and replacement with silicone oil. In all six eyes, MH closure was obtained. None of the eyes developed a rhegmatogenous retinal detachment, proliferative vitreoretinopathy or free graft dislocation. The pre-operative characteristics of all patients are summarized in Table 1.

In eyes with MH, the baseline BCVA was 0.99 ± 0.46 logMAR (Snellen 20/195) (95%CI, 0.51 to 1.47) and the final post-operative BCVA was 1.02 ± 0.23 logMAR (Snellen 20/209) (95%CI, 0.78 to 1.26) (*p*-value 1.00 Wilcoxon Test). BCVA in the control group was 0.05 ± 0.06 logMAR (Snellen 20/22; 95%CI, −0.01 to 0.11).

The mean baseline MH basal diameter was 2443 ± 2956 mm (95%CI, −659 to 5545), the mean linear dimension was 909 ± 635 mm (95%CI, 243 to 1575) and the mean apical inner opening was 1117 ± 648 mm (95%CI, 437 to 1796) (Table 1). All eyes achieved MH closure. The Figure 2 illustrates all the pre-operative and post-operative (6 months) SD-OCTs.

The macular microvascular parameters are summarized in Table 2. The mean post-operative vascular density (VD) in the MH group was 28.1 ± 7.3% (95%CI, 20.4 to 35.8) compared to 20.2 ± 2.9% (95%CI, 12.8 to 27.6) in the control group (*p* < 0.05). The mean post-operative FAZ area of the superficial capillary plexus in the MH group was 131.2 ± 101.2 mm^2^ (95%CI, 24.9 to 237.4) compared to 41.5 ± 10.4 mm^2^ (95%CI, 30.5 to 52.5) in the healthy control group (*p* < 0.05) (Table 2).

## 4. Discussion

Grewal and Mahmoud [13] initially developed ART as a solution to bridge the large gap caused by an MH in a highly myopic patient that had already failed primary surgery, with ILM peeling, and additionally suffered a post-operative retinal detachment. Surprisingly, the patient had a remarkable improvement in visual acuity from 20/200 to 20/80, with a concomitant improvement in distortion, scotoma size and retinal sensitivity. These findings led Grewal and Mahmoud [13] to suggest that, initially, the autologous neurosensory retinal free graft plugs the MH and prevents communication between the vitreous cavity and the subretinal space, allowing the MH to close. Thereafter, it is possible that the neurosensory retinal free graft integrates with the adjacent retinal tissue. Wu et al. [21] proposed using an autologous blood clot to help secure the graft in place. The observation by Tabandeh [20] that vascularization and reperfusion of the graft within 6 weeks of surgery lends further evidence to this hypothesis. A common finding in eyes that undergo ART is a transient hyperreflectivity of the ART graft on SD-OCT. This may represent transient hypoxia of the graft and typically appears during the first post-operative week. In most cases, the hyperreflectivity resolves spontaneously [22].

The macular microvasculature may play a role in the healing process following surgical repair of a MH. OCT-A is an excellent tool to study the macular microvasculature. Pre-operative en face SD-OCT images of MH demonstrate a central hyporeflective area representing the MH that is surrounded by multiple hyporeflective spaces that correspond to cystic areas. The OCT-A shows an enlargement of the FAZ when compared to the fellow uninvolved eye [23]. In eyes with successfully repaired MH, the FAZ is usually smaller than the normal fellow eye [24,25]. Closure of the MH leads to resolution of the cystic spaces, thus causing contraction of the FAZ. As part of the healing process, retinal vessels and tissue replace the cystoid spaces in the parafoveal area. OCT-A (6 months after ART) demonstrates an increase in foveal vessel density and increased parafoveal choriocapillary flow area [23,26]. Given the lack of pre-operative OCT-A images in our patients, we used the healthy fellow eye as a control. In our series, the post-operative vascular density was increased when compared to the unoperated-upon fellow eye, but the FAZ remained enlarged when compared to the fellow healthy eye.

Although we achieved MH closure in all eyes, our patients did not gain significant visual acuity. This is in contrast to most other publications where patients tended to gain vision when the MH closure was achieved [14,22,27,28]. Similarly, Demirel et al. [29] reported a series of eyes with unsatisfactory visual improvement despite MH closure whose FAZ areas were also increased when compared to their control eyes. They speculated that the recovery of the macular microvasculature, in particular the FAZ, following surgical repair may influence post-operative visual acuity. The FAZ area may serve as a biomarker for the foveal neurovascular unit integrity [29]. We suspect that, in our cases, the ART acted more like a scaffold to help bridge the tissue foveal gap and facilitate MH closure. In eyes that experience a significant improvement in vision, ART graft integration and survival is a distinct possibility. The details enabling graft integration and survival remain speculative and may involve ectopic synaptogenesis. Ectopic synaptogenesis, where functional connections between photoreceptors and bipolar and horizontals cells are rebuilt, has been documented in the mammalian retina, albeit not in the human eye [30]. Alternatively, activation of quiescent neural stem cells may also be at play [31]. Tanaka and co-workers [27] noted that, in their series of seven eyes, the retinal microvasculature did not survive in the grafted retina. As a result, the inner nuclear layer became atrophied. In contrast the underlying choroidal blood flow allowed the outer retinal layer to remain intact [27]. Lumi et al. [32] reported the multifocal ERG and microperimetric findings of four patients that underwent ART. In all four eyes, the MH was closed. Ellipsoid zone reconstitution with mild improvement of visual acuity was observed, as well. The multifocal ERG and microperimetric findings suggest that the outer edges but not the central portion of the graft may have become functionally integrated even after almost 2 years of follow-up [32]. Much remains to be learned from this promising technique.

The main limitations of this study were the small number of patients, its retrospective design, the relatively short follow-up, the lack of standardized imaging and the lack of pre-operative OCT-A measurements. In summary, MH closure was achieved in all eyes. Post-operative visual acuity did not improve after ART. Eyes with a closed MH showed a bigger FAZ with a higher VD compared to the fellow healthy eyes.

## Figures and Tables

**Figure 1 jcm-12-02350-f001:**
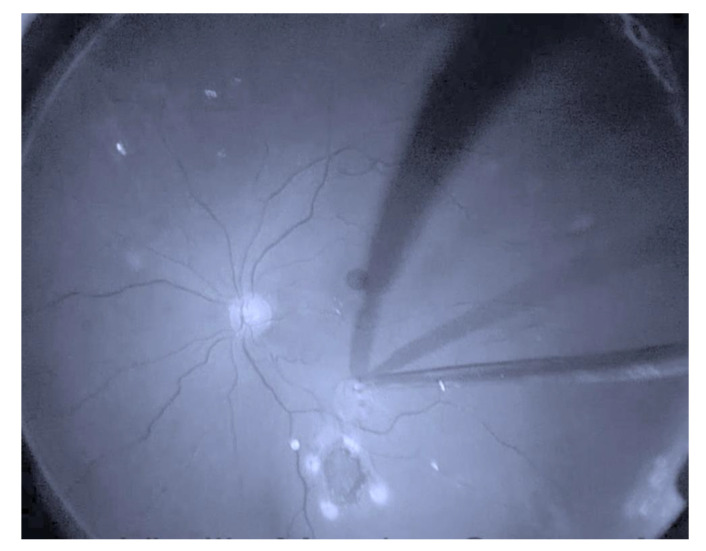
Retinal graft is displaced using ILM (Internal Limiting Membrane) forceps.

**Figure 2 jcm-12-02350-f002:**
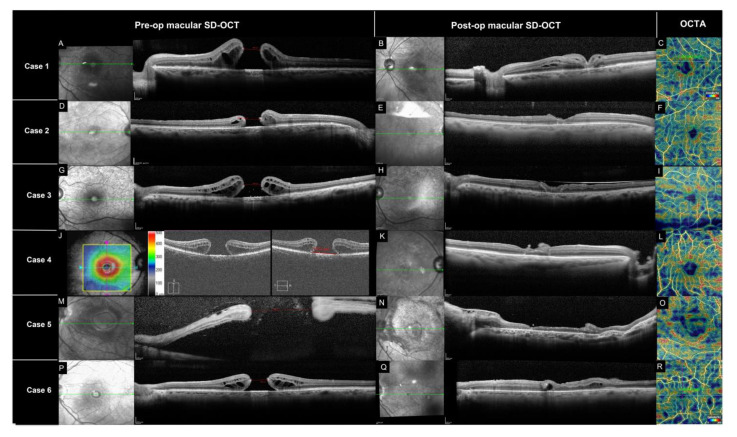
SD-OCT of pre-op and post-op (6 months) MH and ART. OCT-A density map. **Case 1** (**A**–**C**): (**A**) Left eye pre-operative macular SD OCT with MH and intraretinal cyst (Spectralis, Heidelberg Engineering, Heidelberg, Germany). (**B**) Post-operative macular SD-OCT with central ART, presence of some intraretinal cyst. (**C**) Density map (Topcon, Triton, Topcon Healthcare). **Case 2** (**D**–**F**): (**D**) Right eye pre-operative macular SD OCT (Spectralis, Heidelberg Engineering, Heidelberg, Germany). (**E**) Post-operative macular SD-OCT with central ART. (**F**) Density map (Topcon, Triton, Topcon Healthcare). **Case 3** (**G**–**I**): (**G**) Left eye pre-operative macular SD OCT MH with intraretinal cyst (Spectralis, Heidelberg Engineering, Heidelberg, Germany). (**H**) Post-operative macular SD-OCT with ART. (**I**) Density map (Topcon, Triton, Topcon Healthcare). **Case 4** (**J**–**L**): (**J**) Right eye pre-operative macular OCT (Cirrus HD-OCT 4000, Zeiss) 1154 um. (**K**) Post-operative macular SD-OCT with ART. (**L**) Density map (Topcon, Triton, Topcon Healthcare). **Case 5** (**M–O**): (**M**) Left eye pre-operative macular SD OCT, subretinal fluid (Spectralis, Heidelberg Engineering, Heidelberg, Germany). (**N**) Post-operative macular SD-OCT with ART. (**O**) Density map (Topcon, Triton, Topcon Healthcare). **Case 6** (**P**–**R**): (**P**) Left eye pre-operative macular SD OCT and intraretinal cyst (Spectralis, Heidelberg Engineering, Heidelberg, Germany). (**Q**) Post-operative macular SD-OCT with ART and intraretinal cyst. (**R**) Density map (Topcon, Triton, Topcon Healthcare). MLD (minimum lineal dimension), FAZ (foveal avascular zone), MH (macular hole).

**Table 1 jcm-12-02350-t001:** Pre-operative characteristics of eyes undergoing autologous retinal transplant.

Case	Gender	Age	Pre-Op VA (logMAR)	Post-Op VA (logMAR)	BD (mm)	MLD (mm)	AIO (mm)	# of Prior Surgeries
1	M	70	0.6	1	1302	593	726	2
2	F	69	0.69	0.69	698	597	705	1
3	F	67	0.87	0.87	1392	634	818	1
4	F	67	1.77	1.3	1552	851	1229	1
5	M	35	1.3	1.24	8449	2188	2381	0
6	M	74	0.69	1	1266	590	841	1
Mean		63.7	0.99	1.02	2443	909	1117	1

VA = visual acuity; logMAR = logarithm minimum angle of resolution; BD = basal diameter; MLD = minimum lineal dimension; AIO = apical inner opening.

**Table 2 jcm-12-02350-t002:** Comparison of macular microvascular parameters between closed macular hole and the fellow unoperated-upon eye.

Case	Vascular Density Study Eye (%)	Vascular Density Fellow Eye (%)	FAZ Area Study Eye (mm^2^)	FAZ Area Fellow Eye (mm^2^)
1	17.84	17.61	156.4	45.4
2	25.4	19.90	57.0	45.8
3	24.99	32.67	69.1	25.4
4	28.02	22.42	128.1	41.5
5	33.5	12.18	320.0	56.0
6	38.86	16.45	56.3	34.9
Mean	28.10	20.20	131.2	41.5

FAZ = foveal avascular zone.

## Data Availability

Data are available upon request from vmoralesc@mac.com.

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
