# Peer review of "Optical Coherence Tomography Angiography in Macular Holes Autologous Retinal Transplant"

_jcm, 2023, doi:10.3390/jcm12062350_

Round 1

Reviewer 1 Report

This is an interesting descriptive article presenting 6 cases with ART.  For this kind of report, detailed information would be most appreciated.

Major concern is that, although it emphasized on OCT-A analysis on FAZ, it does not show any OCT-A image and the authors do not describe how they evaluated “vascular density”, like they measured over which part, of what area size, etc.  Please describe the detailed protocol in method. It is also appreciated if the authors could share some images of how vascular structure looks like by OCT-A at the closing area with some comments.

The authors report on 6 cases. In Figure 2, why 5 eyes? Since the number of the eyes in this paper is not that large, the presentation of OCT images of all cases are appreciated.

Minor points, but please make it clear when are the post-op VA, OCT, FAZ taken for each case.  Are they of the same post-op months, or different from an eye to eye?

Author Response

We thank for your time and constructive criticism regarding our submitted manuscript. Thanks to your comments our manuscript is better. We hope that we have answered satisfactorily all their queries.

Response to Reviewer 1

Point 1. Major concern is that, although it emphasized on OCT-A analysis on FAZ, it does not show any OCT-A image and the authors do not describe how they evaluated “vascular density”, like they measured over which part, of what area size, etc.  Please describe the detailed protocol in method. It is also appreciated if the authors could share some images of how vascular structure looks like by OCT-A at the closing area with some comments

Response: As suggested by the reviewers we have created Figure 3 where the OCTA images are depicted at 6 months of follow-up. Under Examination Procedures we have expanded the methodology of how we measured the vascular density.

Examination Procedures

At baseline and at each scheduled post-operative visit, each patient underwent a complete ophthalmic examination including best corrected visual acuity (BCVA), slit lamp examination, intraocular pressure (IOP), fundus examination and spectral domain optical coherence tomography (SD-OCT). A volume scan centered on the fovea was performed. The scans were reviewed, manual corrections were performed in case of segmentation errors. Prior to surgery, all patients were staged by SD-OCT (Spectralis, Heidelberg Engineering, Heidelberg, Germany) examinations with measurements of the MH base diameter, the MH minimum linear dimension (width of MH at the narrowest point) and the MH apical inner opening.

Post-operative OCT-A imaging centered on the foveal area (6.0x6.0mm-320x320) with a swept source device (Topcon, Triton, Topcon Healthcare) that uses split-spectrum amplitude-decorrelation was obtained at 6 months post-operatively in both eyes of each patient. The FAZ area was manually measured on the internal edge in the density map and the vascular densities (VD) in percentage (%) were measured with the device's propietary software (Topcon, Triton, Topcon Healthcare) and compared with those of the un-treated fellow eyes. Factors related to macular vascular structural integrity and their associations with postoperative visual outcomes were evaluated.

Point 2: The authors report on 6 cases. In Figure 2, why 5 eyes? Since the number of the eyes in this paper is not that large, the presentation of OCT images of all cases are appreciated.

Response: We have added the OCT images of the missing patient

Point 3: Minor points, but please make it clear when are the post-op VA, OCT, FAZ taken for each case.  Are they of the same post-op months, or different from an eye to eye?

Response: All were taken at 6 months post-op

Reviewer 2 Report

Recurrent macular hole is a difficult problem for retinal surgeirs, this paper introduced a novelty autologus retinal transplantation to repair the macular hole. However, the results presentation should be more to demonstrate the transplant tissue is involved  in process of the macular hole repair.

1, The figure 2 provided 5 patients information, there are 6 patients included in this study. Please provide the 6th patient's SD-OCT figure.

2, The follow-up time and times in this study did not provide. So in figure 2, we don't know when the SD-OCT results get after surgery. In order to demonstrate the autologus retinal tissue play an important role in the macular hole repair, the dynamic SD-OCT, such as 1 week, 1 month, 3 month after transplantation is necessary.

3, The vascular density and FAZ changes should not be compared with the fellow eye. FIrstly, VD and FAZ shold be observed continunously after surgery, then you can find out whether changes occur. Secondly, If the author want to demonstrate the autologus retina contribute to the macular hole closure, some patients using traditional method to treat the macular hole, such as expanded ILM peeling and flap technique, can be included the control group.

Author Response

We thank for your time and constructive criticism regarding our submitted manuscript. Thanks to your comments our manuscript is better. We hope that we have answered satisfactorily all their queries.

Response to Reviewer 2

Point 1. The figure 2 provided 5 patients information, there are 6 patients included in this study. Please provide the 6th patient's SD-OCT figure.  

Response: We have added the OCT images of the missing patient.

Point 2.  The follow-up time and times in this study did not provide. So in figure 2, we don't know when the SD-OCT results get after surgery. In order to demonstrate the autologus retinal tissue play an important role in the macular hole repair, the dynamic SD-OCT, such as 1 week, 1 month, 3 month after transplantation is necessary.

Response: This was a retrospective study so the only available images were at post-op month 6.

Point 3.  The vascular density and FAZ changes should not be compared with the fellow eye. FIrstly, VD and FAZ shold be observed continunously after surgery, then you can find out whether changes occur. Secondly, If the author want to demonstrate the autologus retina contribute to the macular hole closure, some patients using traditional method to treat the macular hole, such as expanded ILM peeling and flap technique, can be included the control group.

Response: With all due respect to the reviewer we disagree that the OCTA parameters should not be compared with the fellow eye. The fellow eye serves as a control for systemic vascular conditions (ie hypertension) that may influence the macular circulation. Unfortunately, as this was a retrospective study, we do not have pre-operative OCTA as mentioned in the limitations of the study.  

As mentioned, 5/6 eyes underwent ART because of prior surgical failure using traditional surgical methods. The fact that the holes closed in of itself demonstrates that ART is a viable alternative in eyes that fail conventional techniques.

Round 2

Reviewer 1 Report

The authors made adequate improvement. Please just include in the legend and text what months are meant by "postoperative."

Author Response

Again, we thank for your time and your comments.

Response to Reviewer 1

The authors made adequate improvement. Please just include in the legend and text what months are meant by "postoperative.

Response: We include on legend and text the time meant as postoperative (6 months).

Reviewer 2 Report

Although the author has reviesed the manusription carefully, the results presentation is not enohgh to demonstrate the role of autologous retinal transplant on macular hole repair. especially there is only one OCT information after surgery. 

Author Response

Response to Reviewer 2

Point 1. Although the author has reviesed the manusription carefully, the results presentation is not enohgh to demonstrate the role of autologous retinal transplant on macular hole repair. especially there is only one OCT information after surgery.

Response: We beg to differ. As we stated "The purpose of the current study is to compare the post-operative macular vascular characteristics in eyes with a MH that underwent ART and the un-treated fellow eye."  We never intended or mentioned that our goal was to demonstrate the role of autologous retinal transplant on macular hole repair. 

In addition we also don't agree that because there is only one post-op OCT there is no useful information that can be learned from it. Obviously if one was postulating a sequence of events then yes, serial OCTs are useful. What can we learn from a post-op OCT? In our series we learned that macular holes were closed following autologous retinal transplant (ART). We also showed that compared to the fellow un-operated eye the FAZ and the VD were higher in the eyes subjected to ART. 
